



# A decade-long hydrographic moored time series near the Drygalski Ice Tongue, Terra Nova Bay, Ross Sea.

Liv Cornelissen[1,2], Sukyoung Yun[3], Jasmin McInerney[1], Brett Grant[1], Fiona Elliott[1,5],
Seung-Tae Yoon[3,4], Christopher J Zappa[6], Won Sang Lee[3], and Craig Stevens[1,2]

[1]Earth Sciences New Zealand, formerly National Institute of Water and Atmospheric Research, Greta Point, Wellington 6021, New Zealand
[2]Department of Physics, University of Auckland, Auckland 1142, New Zealand
[3]Korea Polar Research Institute, Yeonsu-gu, Incheon 21990, Republic of Korea
[4]now at Kyungpook National University, Daegu, South Korea
[5]now at University of Bergen, Norway
[6]Lamont Doherty Earth Observatory, Columbia University, Palisades, NY, USA

**Correspondence:** Liv Cornelissen (liv.cornelissen@niwa.co.nz)

**Abstract.** In this paper we describe a decade-long timeseries of hydrographic mooring observations around the Drygalski Ice Tongue in southern Terra Nova Bay, Antarctica. Unique aspects of the data are that (i) the instruments were placed very close to the Ice Tongue due to its significant influence on the region's ocean and sea ice, and (ii) the upper sensors were positioned relatively close (< 100 m) to the ocean surface compared to typical Antarctic moorings. Starting in December 2014, the moor-

ing array included three locations - the Drygalski Basin, the edge of the Crary Bank and, on the southern side of the ice tongue, in Geikie Inlet. The instruments measure temperature, salinity, pressure, and current velocity. The "DITx" mooring locations were chosen in order to support questions regarding the influence of the Drygalski Ice Tongue on regional ocean processes. The observations are relevant to water mass formation, glaciology and ice melt, sea ice production and decay, ice shelf cavity processes - as well as regional marine ecosystem processes. The data from the instruments show the seasonal cycle along with

interannual variability, as well as a range of singular events. All data can be downloaded from the SEANOE database.

## 1 Introduction

### 1.1 The Drygalski Ice Tongue and Terra Nova Bay

The Drygalski Ice Tongue is presently the largest ice tongue in Antarctica and it forms the southern boundary of Terra Nova Bay (TNB), Western Ross Sea, Antarctica. It is the floating extension of the David Glacier and plays a crucial role in maintaining the Terra Nova Bay polynya by limiting northward-flowing of sea ice along the Victoria Land Coast (Gomez et al. , 2023).

The free floating part of the ice tongue extends ∼80-90 km into the Ross Sea, with a width of ∼20 km (Stevens et al. , 2017) as shown in Figure 1, which makes it Antarctica's largest free-floating glacier (Frezzotti & Mabin , 1994). At the grounding line it is more than 1900 m thick and is ∼150-400 m thick at the tip (Indrigo et al. , 2021; Bianchi et al. , 2001). The tip of the ice tongue extends above the edge of the Crary Bank at a water depth of 600-700 m while the ice tongue is ∼ 200 m thick (Indrigo et al. , 2021). Despite the blocking effect of the Drygalski Ice Tongue on sea ice and surface water masses, Stevens et al. (2024) determined that there is exchange and advection of water masses below the ice tongue.

Terra Nova Bay is bounded to the south by the Drygalski Ice Tongue (DIT) and to the west by the Nansen Ice Shelf, which is fed by the Priestly and Reeves Glacier. A key bathymetric feature, the Drygalski Basin, extends north east into the Drygalski Trough and is believed to be the primary export pathway for High Salinity Shelf Water (HSSW). Along the eastern boundary, the Victoria Land Coast Current carries water and sea ice northward from McMurdo Sound (Stevens et al. , 2017). This flow has been inferred from satellite sea ice drift and surface drifters (Greg Leonard, Antarctic Science Platform's Sea Ice and Carbon Cycle Feedbacks Project) and is reproduced in ocean circulation models such as GLORYS and P-SKRIPS (Malyarenko et al. , 2023), though its vertical structure and depth extent remain poorly constrained by direct observations between McMurdo Sound and Terra Nova Bay. This current could also bring water masses into Terra Nova Bay below the Drygalski Ice Tongue or along the Drygalski Ice Tongue as replenishing the HSSW outflow.

The polynya that occasionally forms in Terra Nova Bay is relatively small in extent when compared to the nearby Ross Sea polynya (Maqueda et al. , 2004) but plays an important role in the global climate system. Between autumn and spring, strong katabatic winds blow off the Nansen Ice Shelf, pushing the sea ice off the coast and enabling heat loss from the relative warm ocean to the cold atmosphere, and producing sea ice and HSSW. Despite its small size in global terms, Terra Nova Bay accounts for approximately 3–4% of Antarctic sea ice production (Tamura et al. , 2016), and up to 10% of Antarctic Bottom Water (AABW). As sea ice makes ship-based observations impossible from late autumn when these important water masses are formed and HSSW is an important water mass that contributes to basal melting of floating glaciers and ice shelves (Nicholls , 1997). Hydrographic moorings, that observe water masses throughout the year are therefore an essential tool to study the oceanographic processes in Terra Nova Bay polynya. While moorings in the central Terra Nova Bay have provided a long term perspective of water mass transformation (MORSea: Marine Observatory in the Ross Sea , 2009; Castagno et al. , 2019), the strong influence of the Drygalski Ice Tongue on the region provided motivation for a mooring array (hereinafter called the "DITx" array) focused around the DIT.

## 1.2 Processes that interact with the Drygalski Ice Tongue

One of the key processes influencing the region are the occasional katabatic-driven polynya events. In such conditions, the fast, cold katabatic winds drive sea ice formation at the ocean surface, which results in the creation of HSSW as a by-product (e.g., Yoon et al. , 2020; Friedrichs et al. , 2022). The brine rejection from the sea ice formation first breaks down the stratification



of the water column in the polynya before HSSW starts to form. While the production of HSSW begins in late Autumn/ early winter, the salinity increase does not increase until September at the bottom in the eastern side of Terra Nova Bay, where DITN is located, and in the Drygalski Basin (Yoon et al. , 2020). The circulation pattern in Terra Nova Bay is described by Yoon et al.

(2020), based on summer CTD observations and three hydrographic moorings between December 2014 and March 2018 and include data from the hydrographic moorings described in this paper (DITN and DITD). They observed a cyclonic circulation in the deeper part of Terra Nova Bay between 400-700 m during the summer, which advects the HSSW created close to the Nansen Ice shelf, which is considered to be the primary location of HSSW production in Terra Nova Bay, towards the DITN, where the salinity increase is observed by the bottom instrument of this mooring. The salinity increases at this depth are ob-

served between September and October, and not at the start of winter when the stratification has broken down. Rusciano et al. (2013) suggested that HSSW is advected from the Nansen Ice Shelf front towards eastern Terra Nova Bay and was confirmed in Yoon et al. (2020).

Terra Nova Bay Ice Shelf Water (TISW) is formed when HSSW interacts with basal melt waters from the Drygalski Ice Tongue

and the Nansen Ice Shelf during active HSSW production.

TISW is characterized by a temperature close to the surface freezing point at -1.94°C and a salinity larger than 34.7 psu (Rusciano et al. , 2013; Yoon et al. , 2020), [fig 6 in (Yoon et al. , 2020)]. The characteristics of TISW observed in summer in Yoon et al. (2020) depends on observation period and the outflow from under the Nansen Ice Shelf and is believed to advect in the same cyclonic circulation as HSSW towards the eastern Terra Nova Bay. TISW is also observed in the eastern Terra Nova

Bay close to Drygalski Ice Tongue by DITN, further study needs to investigate if this TISW is formed under the Nansen Ice Shelf and advected or is formed when HSSW interacts with basal melt waters from the Drygalski Ice Tongue itself. Antarctic Surface Water (ASW) can be found at the subsurface and originates from the melting of the sea ice, it is a fresh and warm (>-1.7 °C, (Friedrichs et al. , 2022)) water mass as it warms through solar radiation during Summer (Rusciano et al. , 2013). As its density is much lower than the water masses formed during winter, it contributes to the restratification of the water column

during the summer months.

### 1.3 Monitoring the water masses around the Drygalski Ice Tongue

The DITx hydrographic moorings distributed around the Drygalski Ice Tongue have been maintained since December 2014 in order to monitor the water mass properties and currents close to the ice tongue. Three moorings were deployed over different

time frames, located around the Drygalski Ice Tongue as shown in Fig. 1. DITN is a long mooring deployed on the northern side of the Drygalski Ice Tongue from December 2014 to the present. It includes instruments near the seafloor, mid-water column, and subsurface, with the shallowest instrument at 75m depth. Its proximity to the ice tongue and near-surface coverage makes it well suited to study subsurface circulation and potential meltwater influence. DITD, a short mooring deployed in February 2017 and still active, is located in the deepest part of the Drygalski Basin. It captures the densest HSSW observed

in the area, offering a valuable reference point for comparing water mass properties with other locations. DITS was a long

mooring positioned south of the Drygalski Ice Tongue in Geikie Inlet and was maintained from February 2017 until January 2020. This paper presents a comprehensive overview of the decadal dataset collected by the DITx array. The data are freely available via SEANOE in netCDF format, organized by mooring, instrument type, and deployment year (Cornelissen et al., 2025). Tables 3, 4, and 5 summarize the mooring configurations, and instrument placements are visualized in Fig. 3.

In addition to these moorings, other relevant datasets in the region include the MORSea Moorings D and L (MORSea: Marine Observatory in the Ross Sea , 2009; Castagno et al. , 2019) and a Lamont-Doherty Earth Observatory (LDEO) mooring (Miller et al. , 2022) located near the front of the Nansen Ice Shelf (Miller et al. , 2022). Since 2013, Argo floats have also contributed valuable data to the Ross Sea, with four deployed in Terra Nova Bay during 2021, 2022, 2024, and 2025 (Argo, 2000).

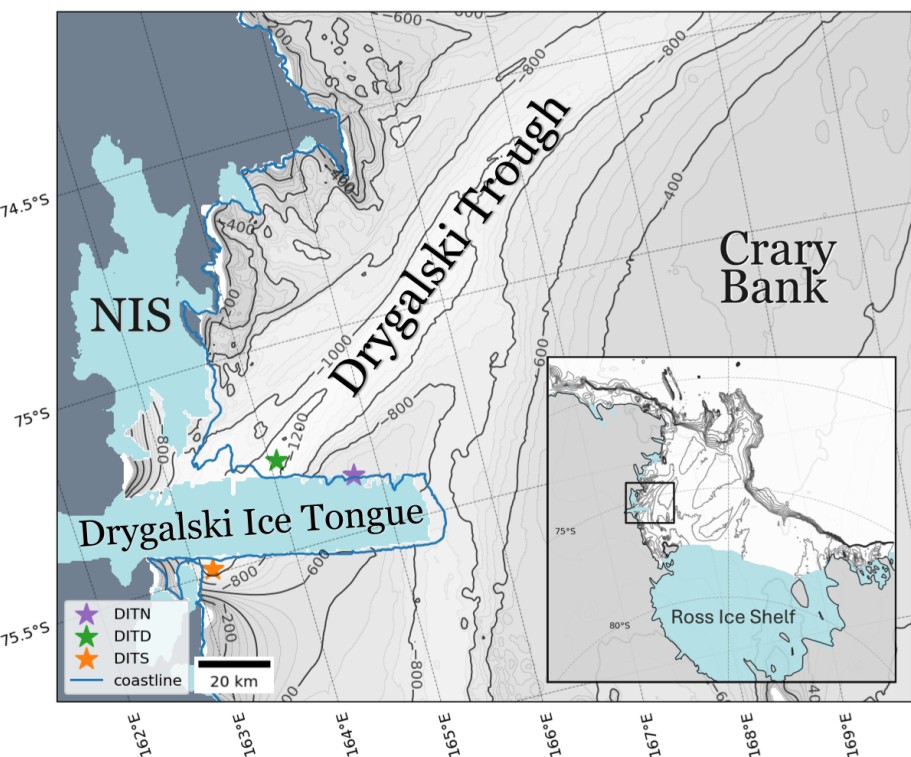

**Figure 1.** Terra Nova Bay (TNB) is located in the western Ross Sea. The Nansen Ice Shelf (NIS) bounds the bay in the west, the Drygalski Ice Tongue bounds the bay on the southern side and the deepest part of Terra Nova Bay is the Drygalski Basin. The eastern boundary of the Drygalski trough is the Crary bank. The DITx mooring locations are marked on the bathymetry of Terra Nova Bay with a Star, the grey contours are the bathymetry at 50 m intervals with solid lines on 200 m intervals. The light blue area represents the floating sea ice, the Nansen Ice shelf and Drygalski Ice Tongue. The dark blue represents the land. DITN mooring was first deployed in December 2014, DITD was first deployed in February 2017 and DITS was deployed from February 2017 until January 2020.



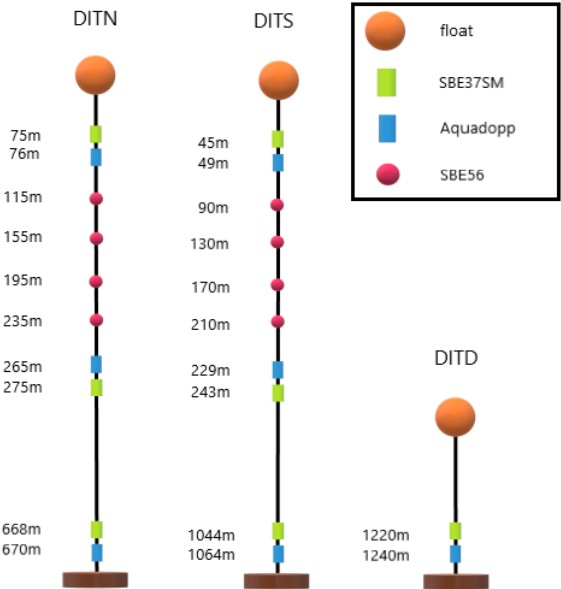

**Figure 2.** A schematic figure of the set up of the three moorings deployed around the Drygalski Ice Tongue in Terra Nova Bay. The DITN has been deployed since December 2014 until present, the DITS between February 2017 and January 2020 and the DITD from February 2017 until present. The Long moorings DITN and DITN have three pairs for CTD (SBE37SMP/SBE37SM) and current meters (Aquadopp) and an sequence of 4 thermistors (SBE56).

## 2 The DITx hydrographic mooring time series

Three hydrographic moorings have been deployed from December 2014 in the Terra Nova Bay polynya equipped with CTD instruments (Sea-Bird Scientific SBE37SM and SBE37SMP, Bellevue), thermistors (Sea-Bird Scientific SBE56, Bellevue) and current meters (RCM9 (Xylem. Inc., Rye Brook) until 2017 and Aquadopp (Nortek, Norway) from 2017. The accuracy of the instruments are shown in Table 1 for the temperature and salinity and Table 2 for the current velocity.

| Instrument | Measurement range | Initial Accuracy |
|---|---|---|
| SBE37SM | -5 to + 35°C | 0.002°C |
| Aquadopp | -4 to + 40°C | 0.1°C /0.01°C |
| SBE56 | -5 to + 45°C | 0.002°C |
| RCM9 | -3.01 to +5.92°C | 0.05°C |
| SBE37SM | 0 to 7 S/m °C | 0.0003 S/m |

**Table 1.** The accuracy of the Thermo-haline instruments based on their manuals. The accuracy of the salinity is not given directly as it is calculated with the conductivity and temperature.





| Instrument | Velocity range | Initial Accuracy | Magnetic resolution |
|---|---|---|---|
| Aquadopp | ±100 cm/s, ±250 cm/s, ±500 cm/s | ±1% of measured value ±0.5 cm/s | 2° for tilt < 30° |
| RCM9 | 0 to 300cm/s | 0.3cm/s | 5° for 0-15°tilt |

**Table 2.** The accuracy of the hydrodynamic instruments based on their manuals. The accuracy of the salinity is not given directly as it is calculated with the conductivity and temperature.

The Drygalski Ice Tongue North mooring (DITN) is located on the northern side of the Drygalski Ice Tongue close to the ice tongue and was first deployed in December 2014. The bottom anchor sits at ∼ 690 m depth. The DITN contains 4 groups of instruments, at the subsurface at ∼75 m depth a MicroCAT SBE37SM and an Aquadopp/RCM9, between 115 m - 235 m depth. 4 SBE 56 thermistors, each 40 m vertical meters apart, a MicroCAT SBE37SM and Aquadopp/RCM9 at ∼275 m depth and a MicroCAT SBE37 and Aquadopp/RCM9 close to the bottom at ∼670 m depth. The RCM9 was used on the moorings: 1412DITN, 1512DITN and 1702DITN, the Aquadopp was first used on mooring 1702DITN, 1702DITD and 1702DITS and all future moorings.

The Drygalski Ice Tongue Deep mooring (DITD) is located slightly north-west of the DITN, in the Drygalski Basin anchored at ∼1250 m depth. This mooring is equipped with 2 instruments near the bottom, a MicroCAT SBE37 and an Aquadopp at around 1240m depth. In 2018, there were 2 mooring groups, both containing a MicroCAT SBE37 and an Aquadopp, at ∼1140 m and ∼1240 m depth.

The Drygalski Ice Tongue South mooring (DITS) was deployed on the southern side of the Drygalski Ice Tongue between February 2017 until January 2020. In the first deployment year between February 2017 and March 2018, it was equipped with a MicroCAT SBE37SM and an RCM9 at ∼190 m depth, 4 thermistors between 230 m and 250 m each 40 m apart, a Micro-CAT SBE37SM and an RCM9 at ∼400 m depth and a MicroCAT SBE37SM and an RCM9 at ∼1100 m depth. The second deployment was equipped with three MicroCAT SBE37SM and an Aquadopp pairs at ∼48 m depth, at ∼230 m depth and at ∼1050 m depth and 4 thermistors between 90 m and 210 m, spaced 40 m apart.

Each mooring was retrieved and redeployed every 1 to 2 years on a voyage, where the data is downloaded and instruments are serviced, batteried, checked and calibrated before being deployed again. The MicroCAT SBE37 measures temperature, salinity (through conductivity) and pressure, the Aquadopp measures temperature, pressure, the direction and speed of the current and heading, pitch and roll, the SBE56 measures the temperature and RCM9 measures the speed and direction of the current.



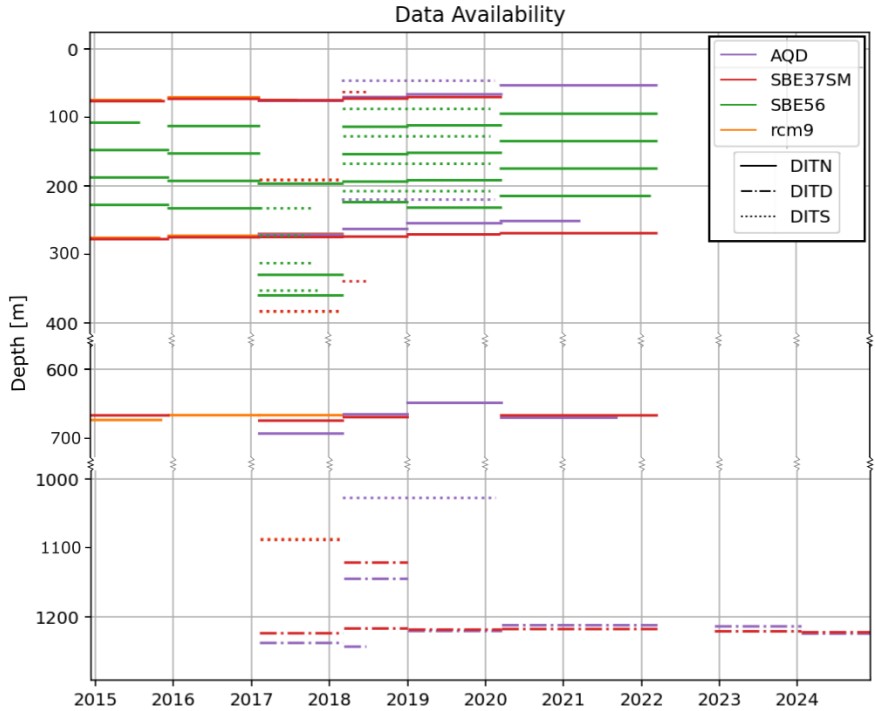

**Figure 3.** The data availability for each of the instruments at their corresponding depth based on the pressure measurements. The solid line are the instruments on the DITN, the dotted line on the DITD and the dot-dashed line on the DITS. The colors correspond with the instrument type, purple for the Aquadopp, red for the ctd SBE37SM, green for the thermistor SBE56 and orange for the current meter RCM9.

## 2.1 Quality control

A number of checks are carried out to provide some assurance on data quality. The data collected by the instruments of the moorings are combined into netcdf files for each of the moorings per instrument type and deployment year. An overview of the available data is summarized in Table 3, 4 and 5.The data is also checked on faulty and missing data, which are described in the corresponding section in the *'Data overview'* section. In some cases, the mooring started the measurement timeseries whilst still on the ship, this resulted in +20° temperatures, and pressure measurements at sea level. This data is removed in before the data processing.

Generally speaking, a CTD profile was performed every time a mooring was deployed or recovered. The salinity measurements of the moored SBE37SM(P) instruments are post-calibrated with the profile measurements using the depth of the moored instruments.





| Instrument | Location | Depth | Type | sn | start [dd/mm/yr] | end [dd/mm/yr] | interval | data quality |
|---|---|---|---|---|---|---|---|---|
| DITN - 1412 | [75 21.612 S, 164 44.971 E] | 75 m | RCM9 | 647 | 12/12/2014 | 20/11/2015 | 1800s | No T data, |
| | | 77 m | SBE37SM | 5675 | 12/12/2014 | 20/11/2015 | 600s | |
| | | 108 m | SBE56 | 2086 | 12/12/2014 | 27/7/2015 | 30s | |
| | | 148 m | SBE56 | 2088 | 12/12/2014 | 10/12/2015 | 30s | |
| | | 188 m | SBE56 | 2089 | 12/12/2014 | 10/12/2015 | 30s | |
| | | 228 m | SBE56 | 2090 | 12/12/2014 | 10/12/2015 | 30s | |
| | | 276 m | RCM9 | 845 | 12/12/2014 | 30/10/2015 | 1800s | |
| | | 278 m | SBE37SM | 7284 | 12/12/2014 | 10/12/2015 | 600S | |
| | | 667 m | SBE37SM | 5838 | 12/12/2014 | 10/12/2015 | 600S | |
| | | 674 m | RCM9 | 847 | 12/12/2014 | 6/11/2015 | 1800S | No T data, |
| DITN - 1512 | [75 21.605 S, 164 44.918 E] | 71 m | RCM9 | 647 | 14/12/2015 | 8/2/2017 | 3600S | |
| | | 73 m | SBE37SM | 5675 | 14/12/2015 | 8/2/2017 | 600S | |
| | | 113 m | SBE56 | 5487 | 14/12/2015 | 8/2/2017 | 30S | |
| | | 153 m | SBE56 | 2088 | 14/12/2015 | 8/2/2017 | 30S | |
| | | 193 m | SBE56 | 2089 | 14/12/2015 | 8/2/2017 | 30S | |
| | | 233 m | SBE56 | 2090 | 14/12/2015 | 8/2/2017 | 30S | |
| | | 273 m | RCM9 | 845 | 14/12/2015 | 8/2/2017 | 3600S | T data faulty, removed |
| | | 275 m | SBE37SM | 7284 | 14/12/2015 | 8/2/2017 | 600S | |
| | | 666 m | SBE37SM | 5838 | 14/12/2015 | 8/2/2017 | 600S | No P recorded |
| | | 667 m | RCM9 | 847 | 14/12/2015 | 8/2/2017 | 3600S | No T data |
| DITN - 1702 | 75 21.646S 164 44.788E | 75 m | RCM9 | 1256 | 9/2/2017 | 1/8/2017 | 3600S | |
| | | 75 m | SBE37SM | 15239 | 9/2/2017 | 6/3/2018 | 120S | |
| | | 76 m | AQD | 13050 | 9/2/2017 | 6/3/2018 | 900S | |
| | | 197 | SBE56 | 4673 | 9/2/2017 | 6/3/2018 | 10S | |
| | | 270 m | RCM9 | 1259 | 9/2/2017 | 6/3/2018 | 3600s | |
| | | 271 m | AQD | 9929 | 9/2/2017 | 6/3/2018 | 900s | |
| | | 275 m | SBE37SM | 15240 | 9/2/2017 | 6/3/2018 | 120s | |
| | | 330 m | SBE56 | 4852 | 9/2/2017 | 6/3/2018 | 10s | |
| | | 360 m | SBE56 | 4854 | 9/2/2017 | 6/3/2018 | 10s | |
| | | 667 m | RCM9 | 342 | 9/2/2017 | 6/3/2018 | 3600 | |
| | | 675 | SBE37SM | 15257 | 9/2/2017 | 6/3/2018 | 120s | |
| | | 694 m | AQD | 9930 | 9/2/2017 | 6/3/2018 | 900s | |

$$\Delta S_{t_1} = S_{CTD_{dep}} - S_{mooring_{t_1}} \tag{1}$$

$$\Delta S_{t_f} = S_{CTD_{rec}} - S_{mooring_{t_f}} \tag{2}$$

$$S(t)_{calibrated} = S(t)_{mooring} + \Delta S_{t_1} + \frac{\Delta S_{t_f} - \Delta S_{t_1}}{N} \cdot t, \tag{3}$$





| Instrument | Location | Depth [m] | Type | sn | start | end | interval [s] | data quality |
|---|---|---|---|---|---|---|---|---|
| DITN - 1803 | [75 21.660 S, 164 44.581 E] | 71 m | AQD | 9929 | 12/3/2018 | 4/1/2019 | 600s | |
| | | 73 m | SBE37SM | 16537 | 12/3/2018 | 4/1/2019 | 300s | |
| | | 114 m | SBE56 | 4855 | 12/3/2018 | 4/1/2019 | 20s | |
| | | 154 m | SBE56 | 4856 | 12/3/2018 | 4/1/2019 | 20s | |
| | | 194 m | SBE56 | 6464 | 12/3/2018 | 4/1/2019 | 20s | |
| | | 224 m | SBE56 | 8011 | 12/3/2018 | 4/1/2019 | 20s | |
| | | 263 m | AQD | 14283 | 12/3/2018 | 4/1/2019 | 600s | |
| | | 274 m | SBE37SM | 16566 | 12/3/2018 | 4/1/2019 | 300s | |
| | | 666 m | AQD | 14372 | 12/3/2018 | 4/1/2019 | 600s | |
| | | 670 m | SBE37SM | 16568 | 12/3/2018 | 4/1/2019 | 300s | |
| DITN - 1901 | [75 20.914 S, 164 48.951 E] | 67 m | AQD | 9929 | 4/1/2019 | 10/9/2019 | 600s | |
| | | 71 m | SBE37SM | 3431 | 7/1/2019 | 13/3/2020 | 300s | |
| | | 112 m | SBE56 | 2098 | 7/1/2019 | 17/3/2020 | 30s | |
| | | 152 m | SBE56 | 2115 | 7/1/2019 | 17/3/2020 | 30s | |
| | | 192 m | SBE56 | 4851 | 7/1/2019 | 17/3/2020 | 30s | |
| | | 232 m | SBE56 | 4857 | 7/1/2019 | 17/3/2020 | 30s | |
| | | 255 m | AQD | 14283 | 4/1/2019 | 17/3/2020 | 600s | |
| | | 271 m | SBE37SM | 2443 | 7/1/2019 | 10/3/2020 | 120s | |
| | | 649 m | AQD | 14372 | 4/1/2019 | 8/9/2019 | 600s | |
| | | 647 m | SBE37SM | 2444 | - | - | - | Faulty data - removed |
| DITN - 2003 | [75 20.958 S, 164 49.021 E] | 53 m | SBE37SM | 2444 | - | - | - | Instrument flooded - no data |
| | | 54 m | AQD | 14292 | 18/3/2020 | 22/7/2020 | 600s | |
| | | 95 m | SBE56 | 4855 | 18/3/2020 | 15/3/2022 | 60s | |
| | | 135 m | SBE56 | 4856 | 18/3/2020 | 15/3/2022 | 60s | |
| | | 175 m | SBE56 | 6464 | 18/3/2020 | 15/3/2022 | 60s | |
| | | 215 m | SBE56 | 8011 | 18/3/2020 | 12/2/2022 | 60s | |
| | | 269 | SBE37SM | 16413 | 18/3/2020 | 15/3/2022 | 300s | No S data |
| | | 252 m | AQD | 14289 | 18/3/2020 | 14/8/2020 | 600s | |
| | | 667 m | SBE37SM | 16417 | 18/3/2020 | 15/3/2022 | 300s | |
| | | 671 m | AQD | 14288 | 18/3/2020 | 24/8/2020 | 600s | |

**Table 3.**

where $S_{CTD_{dep}}$ is the salinity measured by the CTD at depth of the instrument close to the time of deployment , $S_{CTD_{rec}}$ is the salinity measured by the CTD at depth of the instrument close to the time of recovery of the mooring, $S_{mooring_{t_1}}$ is the salinity measured by the mooring just after it is deployed, $S_{mooring_{t_f}}$ is the salinity measured by the mooring just before it is recovered and $N$ is the number of observations made by the mooring. The mooring timeseries is only calibrated if $\Delta S_{t_1}$





| Instrument | Location | Depth [m] | Type | sn | start | end | interval [s] | data quality |
|---|---|---|---|---|---|---|---|---|
| DITD - 1702 | 75 16.542 S, 164 04.038 E | 1225 m | SBE37SM | 15273 | 8/2/2017 | 6/3/2018 | 120s | |
| | | 1239 m | AQD | 13041 | 8/2/2017 | 6/3/2018 | 900s | |
| DITD - 1803 | 75 16.640 S, 164 04.260 E | 1122 m | SBE37SM | 15273 | 10/3/2018 | 3/1/2019 | 60s | |
| | | 1145 m | AQD | 9930 | 10/3/2018 | 3/1/2019 | 600s | |
| | | 1218 m | SBE37SM | 15242 | 10/3/2018 | 3/1/2019 | 180s | |
| | | 1244 m | AQD | 6162 | 10/3/2018 | 22/6/2018 | 600s | |
| DITD - 1901 | 75 16.597 S, 164 04.198 E | 1219 m | SBE37SM | 4840 | 6/1/2019 | 17/3/2020 | 120s | |
| | | 1221 m | AQD | 9930 | 3/1/2019 | 12/3/2020 | 600s | |
| DITD - 2003 | 75 16.681 S, 164 03.616 E | 1213 m | AQD | 14383 | 18/3/2020 | 1/8/2020 | 600s | |
| | | 1218 m | SBE37SM | 15273 | 18/3/2020 | 17/3/2022 | 300s | |
| DITD - 2211 | 75 16.667 S, 164 3.642 E | 1204 m | AQD | 14383 | 10/12/2022 | 21/1/2024 | 600S | |
| | | 1205 m | SBE37SM | 20388 | 10/12/2022 | 21/1/2024 | 300s | |
| DITD - 2401 | 75 16.660 S 164 3.610 E | 1206 m | SBE37SM | 16419 | 21/1/2024 | 8/12/2024 | 300s | |
| | | 1214 m | AQD | 9929 | 21/1/2024 | 8/12/2024 | 600s | |

**Table 4.**

or $\Delta S_{t_f}$ is larger than $0.05 psu$. In case that the instrument stopped recording before recovery, the calibration is done by $S(t)_{calibrated} = S(t)_{mooring} + \Delta S_{t_1}$.

## 3   Oceanographic data

This section lists all the available data measured by the instruments on the DITx moorings in Terra Nova Bay. The mooring is referred to as [mooring][deployment year][deployment month], for example DITN1412. Details about the measurements are

described per variable in the corresponding subsections, including the depth and period where observations are absent or need to be treated with caution due to faulty instruments or empty batteries. The observational daily climatologies are shown for the salinity, temperature, density and speed for the DITN and DITD mooring. The climatologies are not shown for the observations of DITS as this mooring was only maintained for 3 years, with instruments at varying depths.

### 3.1   CTD results

Pressure is recorded by the Aquadopp and the MicroCAT, SBEs; the measurements per depth are plotted in Figure 4. The depth at which the variables are measured, as described in Table 3, 4 and 5, are the mean values of the pressure-derived depth values. The mean depth for instruments that do not record the pressure is calculated by the distance offset from an instrument that does measure the pressure.





The pressure also varies between deployments, as the depth varies due to the exact location of the mooring anchor, therefore the depth of the mooring anchor changes slightly each time the mooring is retrieved and deployed again. The pressure data shows bigger spikes in the instruments closer to the surface compared to the instruments closer to the bottom. These spikes are not removed from the data as they are caused by a increase in depth when the mooring line gets pulled down due to drag imposed by currents.

All SBE37SM, SBE57 and Aquadopp instruments measure the temperature. The depths of these instrument per mooring can be found in Tables 3, 4 and 5. Figure 5 shows the temperatures for each of the instruments. The accuracy of the Sea-Bird instrument are significantly better ($0.002°C$) than the Aquadopp ($0.01°C$) and the RCM9 ($0.05°C$) as shown in Table 1

The temperatures measured in the top 400 m, show a clear seasonal signal, decreasing with depth. The temperature measured by the DITN near the surface at $\sim 75$ m depth increases from November, peaking at the end of February, it decreases during the Autumn months from March until June and during the winter months, the temperatures are $\sim -1.9°C$ and show the smallest

| Instrument | Location | Depth [m] | Type | sn | start | end | interval [s] | data quality |
|---|---|---|---|---|---|---|---|---|
| DITS - 1702 | [75 29.305 S, 163 10.461 E] | 191 m | RCM9 | 647 | 12/2/2017 | 8/3/2018 | 3600s | |
| | | 192 m | SBE37SM | 7227 | 12/2/2017 | 8/3/2018 | 300s | |
| | | 233 m | SBE56 | 4855 | 12/2/2017 | 24/10/2017 | 10s | |
| | | 273 m | SBE56 | 4856 | 12/2/2017 | 2/10/2017 | 10s | |
| | | 313 m | SBE56 | 4857 | 12/2/2017 | 21/10/2017 | 10s | |
| | | 353 m | SBE56 | 6464 | 12/2/2017 | 10/11/2017 | 10s | |
| | | 383 m | SBE37SM | 1627 | 12/2/2017 | 8/3/2018 | 300s | faulty P & S data removed |
| | | 384 m | RCM9 | 845 | 12/2/2017 | 8/3/2018 | 3600s | |
| | | 1087 m | RCM9 | 847 | 12/2/2017 | 8/3/2018 | 3600s | |
| | | 1088 m | SBE37SM | 7284 | 12/2/2017 | 8/3/2018 | 600s | |
| DITS - 1803 | [75 29.313 S, 163 12.251 E] | 47 m | AQD | 14288 | 4/3/2018 | 2020-02-19 | 600s | |
| | | 63 m | SBE37SM | 15239 | 10/3/2018 | 14/6/2018 | 20s | |
| | | 88 m | SBE56 | 4673 | 9/3/2018 | 28/1/2020 | 20s | |
| | | 128 m | SBE56 | 2089 | 9/3/2018 | 28/1/2020 | 20s | |
| | | 168 m | SBE56 | 4852 | 9/3/2018 | 28/1/2020 | 20s | |
| | | 208 m | SBE56 | 4854 | 9/3/2018 | 28/1/2020 | 20s | |
| | | 220 m | AQD | 14289 | 4/3/2018 | 28/1/2020 | 600s | |
| | | 339 m | SBE37SM | 15240 | 10/3/2018 | 18/6/2018 | 20s | |
| | | 1027 m | AQD | 14290 | 4/3/2018 | 28/1/2020 | 600s | |
| | | 1458 m | SBE37SM | 15257 | 10/3/2018 | 18/6/2018 | 20s | |

**Table 5.**





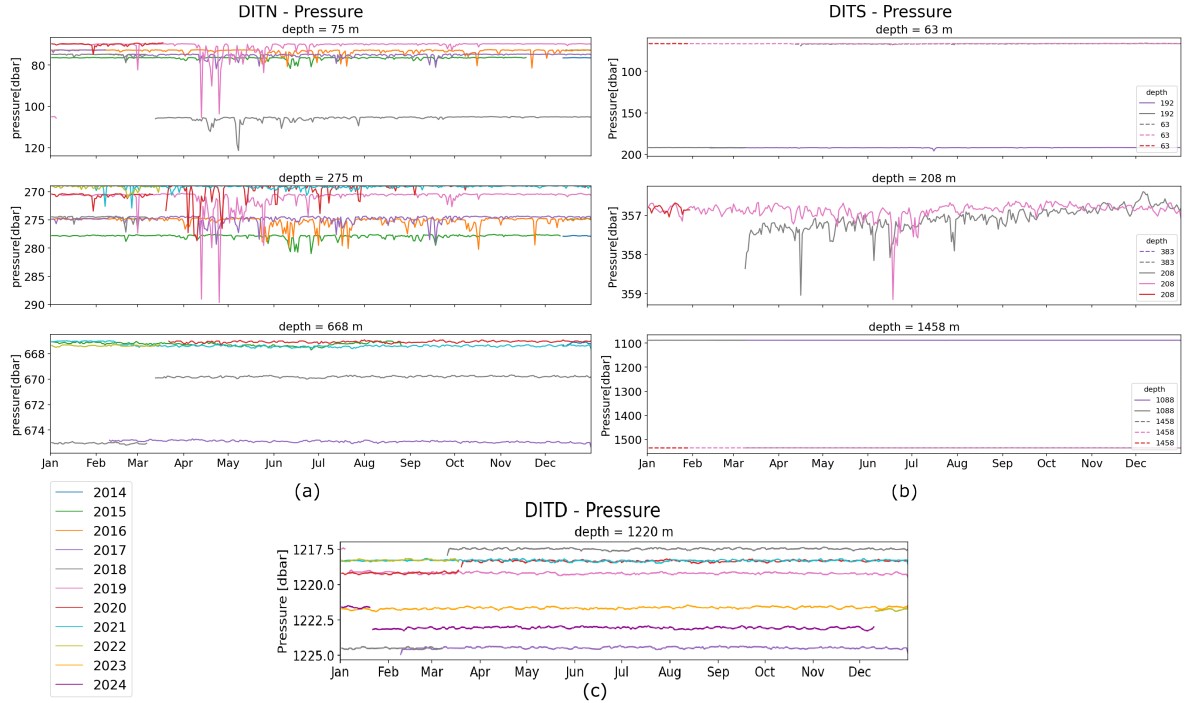

**Figure 4.** The pressure measured by the (a) DITN at $\sim 75$ m, $\sim 275$ m and $\sim 668$ m depth, (b) by the DITS at $\sim 48$ m, $\sim 192$ m, $\sim 220$ m, $\sim 1027$ m and $\sim 1088$ m depth and (c) by the DITD $\sim 1220$ m depth.

variability at this mooring. In the Geiki inlet where the DITS is located, the temperature in the top 100 m increases from the end of December, peak in February and decreases until April where temperatures show low variability throughout the winter
and spring. Between $\sim 200 - 400$ m depth at the DITN and DITS site, the temperature does not increase until late February, during April the highest temperatures of $\sim -1.2$ to $-1.7°$C are recorded and they start to drop again during May and June. The winter period starts in July at the DITN site and in May at the DITS site with temperatures $\sim -1.9°$C with super-cooled water reaching $< -2.0°$C. Near the bottom temperatures range between $-1.87°$C and $-1.91°$C, there is a decrease between July and November at the DITD and DITN location and no clear seasonal signal at the DITS site. There is a strong temperature increase
between end of April and start of May measured by the SBE37SM DITS1803_15239 at 63 m depth and the SBE56 thermistor DITS1803_4854 at 208 m depth. Because this signal is measure by two different instruments at two different depths, it is not removed from the final dataset, but should be treated with caution. Sudden large temperature changes (spikes) that last longer than 1 hr are not removed from the dataset as the polynya is a highly active environment and sudden temperature changes can occur.

The temperatures are measured correctly for most of the instruments and moorings, the measurements that need to be treated with caution, are absent or ignored due to faulty instruments are presented in Table 3, 4 & 5. Aquadopp c1901DITN_9929 at 71

m depth shows a sudden jump in temperature on 22/8/2019. At the same time as the sudden temperature increase, the pressure
increases to 90 dbar at 15:30 and decreases again to 68 dbar at 16:15, while the temperature does not go back to the values
before the sudden increase. This could indicate that the instrument got hit by an iceberg and possibly broke the temperature
sensor. The temperature data after 22/8/2019T15:30 is removed in the final dataset.

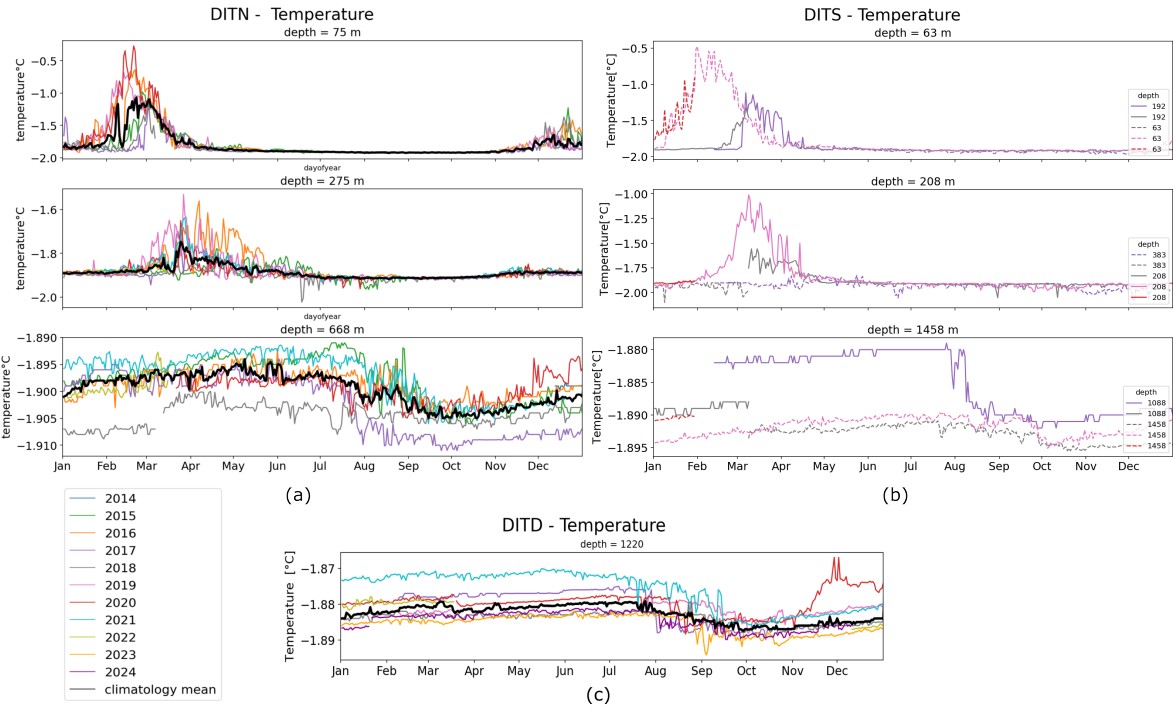

**Figure 5.** The temperature measured (a) by the DITN at $\sim$ 75 m, $\sim$ 275 m and $\sim$ 668 m depth, (b) by the DITS at $\sim$ 48 m, $\sim$ 63 m, $\sim$ 88 m
$\sim$ 192 m, $\sim$ 208 m, $\sim$ 228 m, $\sim$ 383 m, $\sim$ 1064 m, $\sim$ 1088 m and $\sim$ 1458 m depth and (c) by the DITD $\sim$ 1220 m depth. The observational
climatology is represented by the black line for DITN and DITD. DITS does not have enough observations to compute a climatology.

The salinity is measured indirectly by the SBE37SM using the conductivity and temperature in PSU. Near the surface
at $\sim 75m$ depth, measured by the DITN, the salinity decreases in the summer from the middle of January from 34.9psu to
34.2psu. At $\sim 275m$ depth, the salinity oscillates with the highest salinity in September and the lowest salinity in May, close
to the bottom, there is no obvious seasonal signal. In the Drygalski Basin, measured by the DITD, the salinity decreases slowly
from the end of Autumn, with a sharp increase between September and October. The SBE37SM instruments on the DITS1803
mooring, measuring the salinity either did not record any data or stopped early. The salinity measurements are plotted in Figure
6.


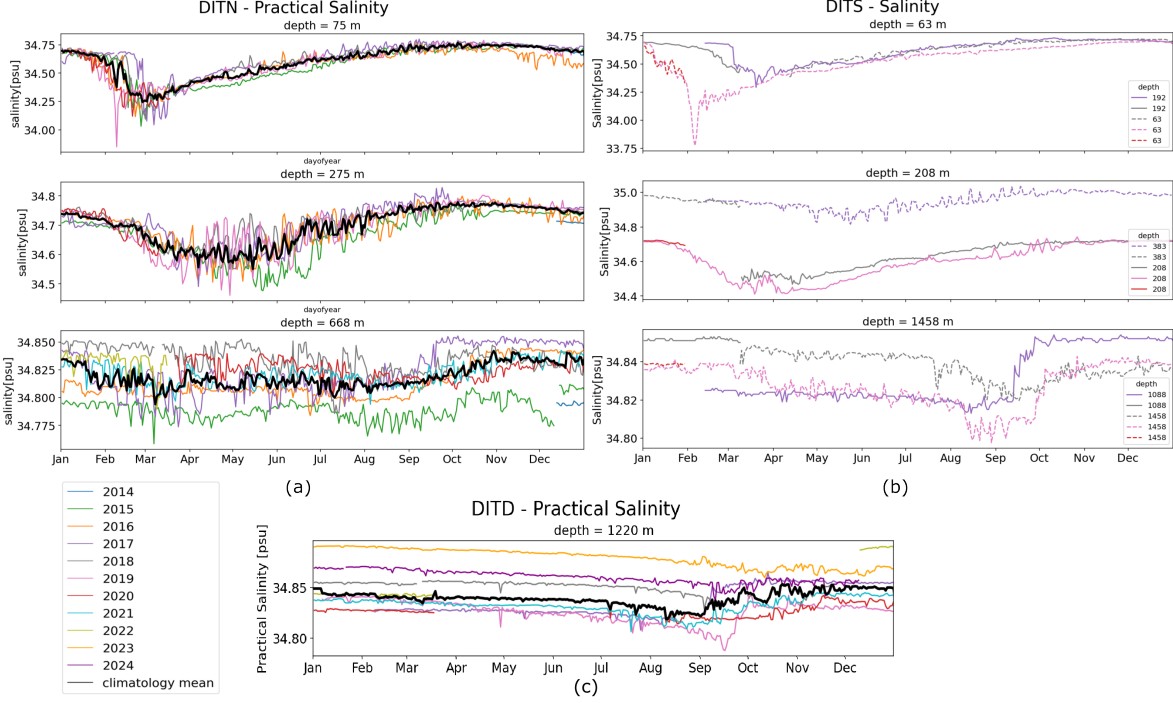

**Figure 6.** The salinity measured (a) by the DITN at $\sim 75$ m, $\sim 275$ m and $\sim 668$ m depth, (b) by the DITS at $\sim 63$ m, $\sim 192$ m, $\sim 208$ m, $\sim 1088$ m and $\sim 1458$ m depth and (c) by the DITD $\sim 1220$ m depth . The observational climatology is represented by the black line for DITN and DITD. DITS does not have enough observations to compute a climatology.

The density is calculated for DITN at 75, 275 & 668 m depth, on the DITD at 1220 m and during DITD1803 also at 1120 m and on the DITS1702 at 190, 390 and 1090 m depth and on DITS1803 at 48, 240 and 1050 m depth. The MicroCAT did not record any density data from the DITN1902. Also there is no density data recorded on the 15/12 mooring at 275 and 668 m depth. The missing density data can be calculated using the TEOS10 function with the salinity, temperature and pressure. The calculated density on the DITN1902 mooring at 668 m depth, is faulty as it is calculated with the salinity and temperature, which was not recorded correctly. The same applies to the DITS1803 mooring data at 48 and 240 m depth.

### 3.1.1 Water masses

We observe the following water masses In Terra Nova Bay near the Drygalski Ice Tongue (DIT), High Salinity Shelf Water (HSSW), defined as $\theta_\rho > 1028.00 kg m^{-3}$, Terra Nova Bay ice shelf water (TISW) where $\theta_T < -1.94°C$ and Antarctic Surface Water (ASW) $\theta_T > -1.7°C$ (Yoon et al. , 2020; Friedrichs et al. , 2022). The TS-diagrams for the three depths that measure salinity in DITN and at DITD are shown in Figure 7.

The variability of the top half of the water column, up to 275 m depth, is forced by surface processes. The temperature and





salinity response in the mid water-column is shifted forward in time compared to the subsurface; in spring, we first observe a
freshening and increase in temperature at 75 m depth while fresher and warmer water masses don't reach deeper within the
water at 275 m depth column until summer. In Autumn, the subsurface also cools and increases in salinity before the mid
water column. The mixed layer depth rapidly increases between March and May and is assumed to be homogeneously mixed
throughout the winter mass (Yoon et al. , 2020), when HSSW is formed through sea ice formation during katabatic wind events.
However, the HSSW formed in the eastern Terra Nova Bay does not convect all the way to the bottom, as the salinity observed
in the top half of the water column is fresher than observed at the bottom instrument of DITN and the annual salinity increase
is observed much later, between September and October, compared to the top half of the water column, as shown in Figure
6(a) and 6(b). Yoon et al.  (2020) found a cyclonic circulation between 400 - 700 m depth in CTD and mooring observations
(DITN, DITD and Mooring D) that advects HSSW formed, in front of the Nansen Ice Shelf, to the bottom instrument of DITN
and the Drygalski Basin.

TISW is observed by DITN in the top half of the water column, where the Drygalski Ice Tongue extends to ∼200-400 m depth.
In the subsurface at 75 m depth, TISW is observed during winter and spring, while at 275 m depth it is observed between
summer and autumn, but with large interannual variability. This is in contrast to the Terra Nova Bay wide CTD observations
in Yoon et al.  (2020) that observed TISW between 300-600 m depth, however, they did find that the presence of TISW was
dependent on the observation period. DITN is located close to the DIT, and the TISW observed by this mooring likely origi-
nates from water masses mixed with basal melt water from the ice tongue. Further study is needed to confirm this, as Yoon et
al.  (2020) found that TISW formed under the Nansen Ice Shelf is advected in a cyclonic circulation similar to the HSSW.



**Figure 7.** The absolute salinity versus the conserved temperature per season for each depth, colored by year. The daily climatology over the observed years is plotted in black. Summer is defined as the period from Jan-1 until Apr-1, Autumn from Apr-1 until Jul-1, Winter from Jul-1 until Oct-1 and Spring from Oct-1 until Jan-1. The grey shading defines the HSSW ($\theta_\rho > 1028[kg/m^3]$), the isotherms define the ISW ($\theta_T < -1.94[^\circ C]$) and AWS ($\theta_T > -1.7[^\circ C]$). The densities are plotted as contour lines.

## 3.2 Current meter results

The speed and direction of the current is measured with the RCM9 and Aquadopp instruments. On the DITN at 74, 76, 265, 274, 670 & 672 m depth. The RCM9 was installed on the DITN1412 & DITN1512 mooring at 74, 274 & 272 m depth and DITN1702 mooring at 74, 274 & 342 m depth and the Aquadopp at 76,265 & 670m depth from the DITN1702 mooring onwards. The DITD measures the currents with the Aquadopp at 1240 m depth. The currents are measured on the DITS1702 at



190, 390 and 1090 m depth with the RCM9 and on the DITS1803 50, 230 and 1060 m depth with the Aquadopp.

Figure 8 shows the hourly means of the current speed for each of the moorings. The currents are corrected for the magnetic declination at the position of the mooring.

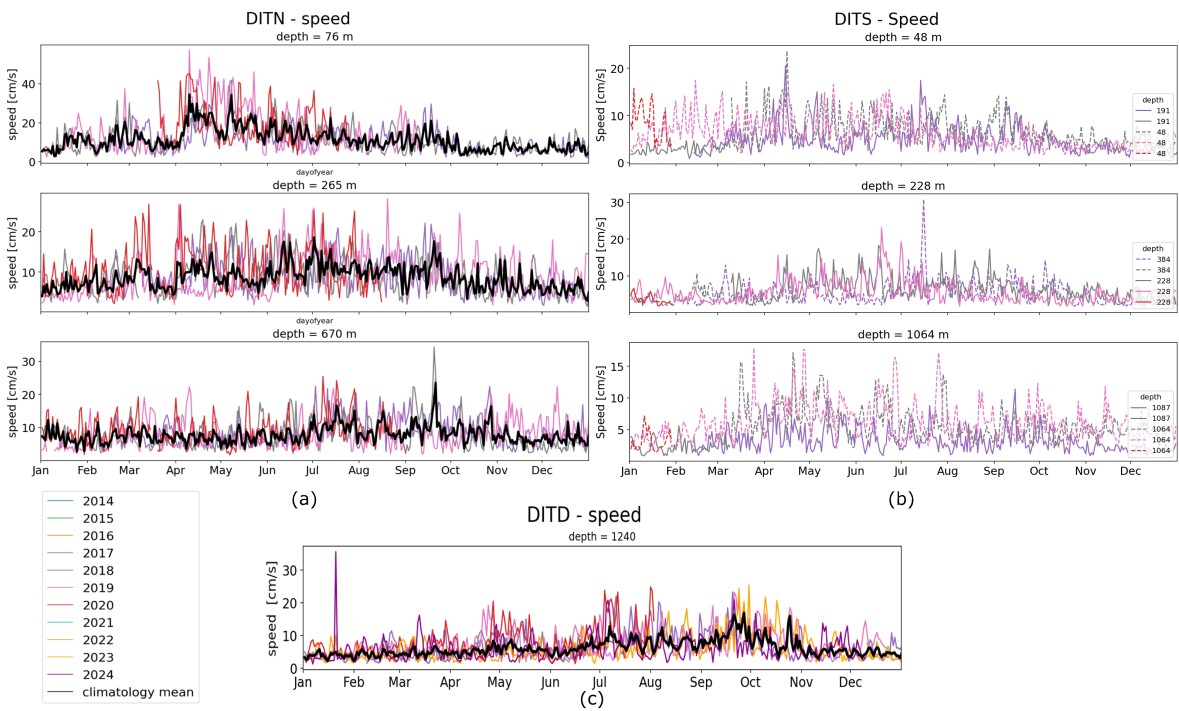

**Figure 8.** The current speed measured (a) by the DITN at ∼ 76 m, ∼ 265 m, and ∼ 670 m depth, (b) by the DITS at ∼ 48 m, ∼ 191 m, ∼ 228 m, ∼ 384 m, ∼ 1064 m and ∼ 1087 m depth and (c) by the DITD ∼ 1240 m depth. The observational climatology is represented by the black line for DITN and DITD. DITS does not have enough observations to compute a climatology.

The prevailing direction of the flow measured by DITN is westerly near the surface at ∼ 75 m depth and in the middle of the
245 water column at ∼ 275 m depth. At the bottom at a depth of 670 meters, the current direction is very variable with a slight south westerly tendency. The prevailing direction of the flow measured in DITD is westward, ranging between north west and south west or in north east direction. The bottom instrument of DITN and DITD show the largest variability, while the subsurface shows the smallest variability. This preferred westerly flow near the surface corresponds with the deflection of the Victoria Land Coastal current deflection observed in satellite data [(Moctezuma-Flores et al. , 2017)].

The heading, Roll and pitch are measured with the Aquadopp on the DITN1702 onwards at 76, 265 & 670 m depth, at 1145 & 1240 m on the DITD moorings and at 50, 230 and 1070 m on DITS1803. The heading is the rotation along the z-axis, the

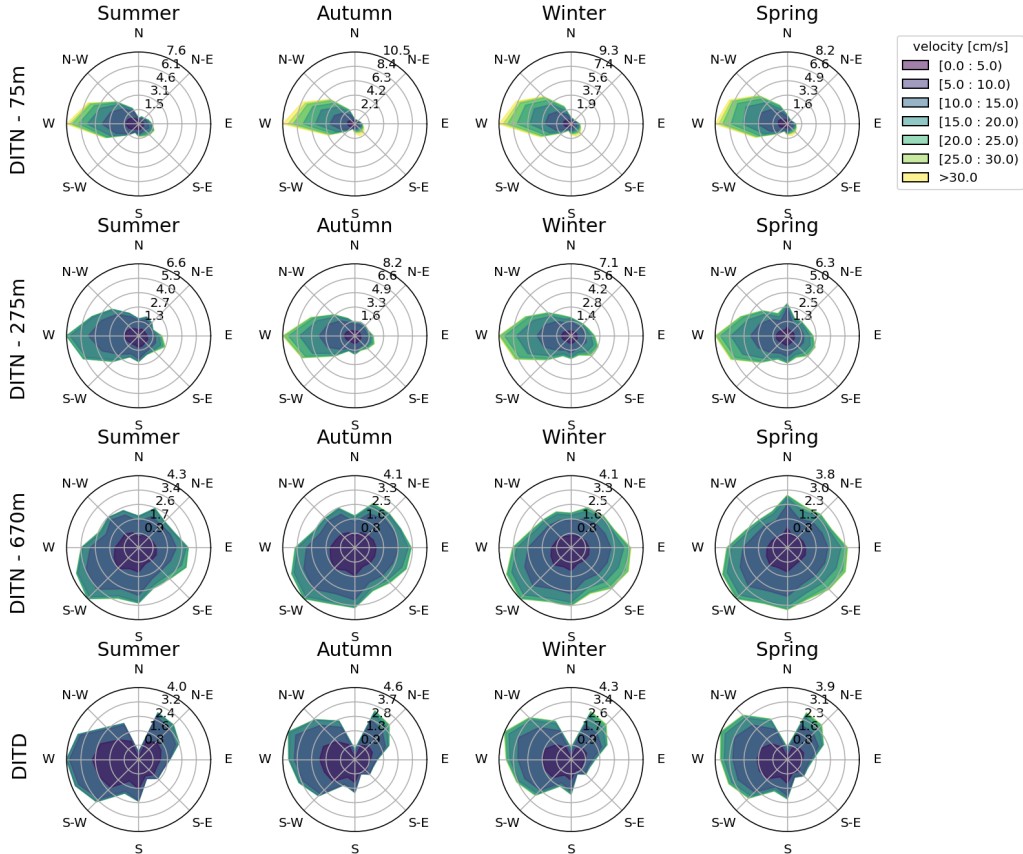

**Figure 9.** The currents measured by the three DITN aquadopps and the DITD are plotted per instrument, per season. The contours represent the direction and speed of the currents of the total observed period. The radius of the bar represents the frequency of the occurrence.

pitch along the x-axis and the Roll along the y-axis. The heading, Roll and pitch data can be used to study the movement of the mooring. The set-up, lengths between the instrument connections and length of the chains of the mooring can explain the variability in between mooring deployments.

## 4 Discussion and Conclusions

This paper shows the data collected by three mooring around the Drygalski Ice Tongue in Terra Nova Bay between December 2014 and December 2024. The Drygalski Ice Tongue is a prominent feature in Terra Nova Bay and blocks the inflow of sea ice from the south, enabling it to operate as a polynya during the colder months of the year. The water masses formed in Terra Nova Bay influence the Drygalski Ice Tongue and changes to these water masses can affect the stability and existence of this ice tongue, as warmer oceans could lead to the break off of the ice tongue. The long-term observations help with monitoring the water masses close to the Drygalski Ice Tongue and can detect any changes that might affect the stability of the ice tongue.



As the Drygalski Ice Tongue also extends into the water column, it also affects the circulation and advection of water masses within Terra Nova Bay and the exchange below the ice tongue (Stevens et al. , 2024). A modelling study that represented
a polynya similar to Terra Nova Bay done by Xu et al. (2023), found that an ice tongue in a coastal polynya changes the circulation patterns compared to polynya that is not bound by an ice tongue. They observed a surface flow along the ice tongue towards the ice shelf front, which corresponds with the westerly flow observed at the subsurface instrument of DITN. Therefore this data set will increase our understanding how the Drygalski Ice Tongue affects seasonal and interannual variability of the circulation patterns close to the ice tongue.

## 4.1 Outlook and uses of data

The described hydrographic mooring timeseries, presented in this paper, are a valuable tool to increase our understanding of the highly dynamical polynya system. Comparing this data set with other observations in Terra Nova Bay, helps with the horizontal extend of the polynya dynamics and variability. The length of the observations closer to the surface, can give insight in the
275 drivers of and interannual variability of water mass formation processes. This long-term dataset allows for the identification of key forcing mechanisms, such as atmospheric conditions, sea ice dynamics, and ocean circulation patterns, that influence the timing, amount, and properties of HSSW and how the Drygalski Ice Tongue itself affects this. We hope determine the origin of water masses observed near the Drygalski Ice Tongue during the winter months when HSSW is formed and bay-wide CTD observations are impossible using idealised and regional models.

The datasets from DITN and DITS are also valuable for biological applications. Both DITN and DITS include instruments located within the euphotic zone, allowing for the analysis of the mixed layer depth and its variability. Due to close proximity of DITN and DITS to the Drygalski Ice Tongue, these datasets offer an opportunity to investigate oceanographic conditions that may influence the stability and evolution of the ice tongue. These moorings can give an in-situ oceanographic perspective of the
285 changes observed with radar and satellite data. When combined with radar and satellite imagery, the in situ data provide a complementary oceanographic perspective, enhancing our understanding of ice-ocean interactions and enabling interdisciplinary studies linking ocean physics, glaciology, and marine biology.

*Code and data availability.* This paper provides a detailed description of the temporal coverage of the dataset, which consists of a near-continuous high-temporal-resolution time series of currents, temperature, and salinity from December 2014 to December 2024. The method-
290 ology adopted for settings, data recording, and quality control ensures the dataset's compliance and consistency. Although the dataset currently ends in December 2024, monitoring activities are ongoing. Future data from this hydrographic timeseries will be added to an updated version of the repository as future moorings are recovered. All coding used to make the datafiles are uploaded to Github: https://github.com/Livcornelissen/DIT_mooringdata/tree/main. The data of the three hydrographic moorings are uploaded as netcdf files to SEANOE in DOI:10.17882/102640 (Cornelissen et al. , 2025)



*Author contributions.* The dataset was assembled and checked by LC. JMc, BG, FE, CJZ, SY, STY and LC led the deployment and recovery of the moorings on various voyages with the assistance of the *Araon* deck crew. The timeseries was initiated by CS, WSL and CJZ. LC wrote up the manuscript and all other co-authors helped improve it.

*Competing interests.* The authors declare that they have no conflict of interest.

*Acknowledgements.* The authors wish to thank the Korea Polar Research Institute (KOPRI), the New Zealand Antarctic Research Institute,
the N.Z. Antarctic Science Platform and Antarctica New Zealand for support. We especially thank the crew of the IBRV *Araon*. The mooring instruments are provided by the NIWA Capex Program. This paper forms a contribution to the N.Z. Antarctic Science Platform (ANTA1801). We thank Gary Wilson and Richard Levy for their support in initiating this work. In addition, we acknowledge Pierpaolo Falco and Pasquale Castagno who assisted in recovery of the DITS mooring with the *Laura Bassi*. SY and WSL are supported by the Korea Institute of Marine Science & Technology Promotion (KIMST) funded by the Ministry of Oceans and Fisheries (RS-2023-00256677; PM25020).



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
