# Peer review of "A decade-long hydrographic moored time series near the Drygalski Ice Tongue, Terra Nova Bay, Ross Sea."

_Earth System Science Data, 2025_

## Referee Comment (RC1)

**Referee Report for ESSD Manuscript: essd-2025-540**

**Summary**

This manuscript presents a valuable decade-long mooring dataset collected in Terra Nova Bay (Western Ross Sea) near the Drygalski Ice Tongue. Such a long-term time series in this area is very important for understanding circulation, shelf processes, dense water formation, and their variability.

The study provides a high-quality time series from a scarcely observed Antarctic region, with clear structure, strong relevance for model validation and process studies, and commendable effort toward open data accessibility

The paper is generally well written and the dataset clearly worth publishing in Earth System Science Data. However, some methodological and interpretative issues should be addressed before the dataset can be considered fully transparent, reproducible, and ready for broad community use. In particular, the calibration procedures, uncertainty estimates, and definitions of key thermodynamic variables require more rigorous documentation.

**Major Comments**

**1. Instrument calibration and quality control**

The calibration and QC procedures are too briefly summarized. The authors should provide a detailed account of the calibration steps for each instrument (laboratory and in-situ), specify the reference CTD profiles used, describe the evaluation of sensor drift, and justify the  $\pm 0.05$  psu criterion for applying corrections.

A dedicated subsection "Instrument calibration and intercomparison" would strengthen reproducibility.

Use of the term climatology

The manuscript refers to an "observational climatology" based on approximately ten years of data. As a 30-year baseline is normally required for a true climatology, the terminology is potentially misleading. Please use expressions such as multi-year mean, decadal mean, or provisional climatology, and clarify the exact temporal coverage.

**2. Quantification of measurement uncertainties**

Instrument accuracies are reported, but there is no propagation of errors to obtain overall uncertainty in temperature, salinity, and current velocity. The paper would benefit from a quantitative uncertainty assessment, including calibration errors, sensor drift, and pressure variations. Confidence intervals or shaded uncertainty bands in key figures would communicate data reliability to users.

**3. Depth assignment and mooring motion**

Depth determination across deployments is not described in sufficient detail. The authors should explain how mean depths were derived, whether time-varying pressure data were used, and how mooring motion was accounted for when merging consecutive deployments. Providing pressure time-series or a summary table of nominal versus mean depths would improve transparency.

**4. Lack of tidal or spectral decomposition**

Separating tidal/inertial from sub-inertial variability might be useful. The manuscript should include at least a basic spectral analysis of the velocity to point the most energetic component of the current field, and include a brief discussion of how high-frequency variability may influence (or not) the presented means and anomalies.

**5. Thermodynamic variable definitions**

It is sometimes unclear whether the reported temperature refers to in-situ, potential, or conservative temperature. The authors should ensure consistent use of TEOS-10 terminology and explicitly state which variables (SA, CT, density) are used in each figure and calculation. Even density indicated as  $\Theta_D$  is pretty unusual (at least to me)

**6. Conclusions somewhat over-generalized**

Some of the final statements about implications for the stability of the Drygalski Ice Tongue and regional climate significance go beyond what the single-site dataset can substantiate. The conclusions should be more cautious and distinguish clearly between direct observational evidence and broader speculative implications.

**Minor Comments**

- Ensure consistency of units (prefer g kg-1 for salinity, °C for temperature).
- Correct small typographical errors and check that instrument names and model numbers are consistent across text and tables.
- Clarify figure captions, especially for T–S plots and "daily climatology" curves, and specify whether density is potential or in-situ.

I look forward to seeing this dataset published after careful revision.